# *Gynostemma pentaphyllum* Hydrodistillate and Its Major Component Damulin B Promote Hair Growth-Inducing Properties In Vivo and In Vitro via the Wnt/β-Catenin Pathway in Dermal Papilla Cells

**DOI:** 10.3390/nu16070985

**Published:** 2024-03-28

**Authors:** Lochana Kovale, Seoyeon Lee, Minhyeok Song, Jihyun Lee, Hyeong Jig Son, Young Kwan Sung, Mi Hee Kwack, Wonchae Choe, Insug Kang, Sung Soo Kim, Joohun Ha

**Affiliations:** 1Department of Biochemistry and Molecular Biology, Graduate School, College of Medicine, Kyung Hee University, Seoul 02447, Republic of Korea; kovlelochana@gmail.com (L.K.); wasdde567777@naver.com (S.L.); thdalsgur77@gmail.com (M.S.); thsgudwlr127@naver.com (H.J.S.); wchoe@khu.ac.kr (W.C.); iskang@khu.ac.kr (I.K.); sgskim@khu.ac.kr (S.S.K.); 2Easy Hydrogen Corporation, Jeju City 63196, Republic of Korea; easyhydrogen@gmail.com; 3Department of Immunology, School of Medicine, Kyungpook National University, Daegu 41944, Republic of Korea; ysung@knu.ac.kr (Y.K.S.); go3004@khu.ac.kr (M.H.K.)

**Keywords:** alopecia, Wnt/β-catenin, *Gynostemma pentaphyllum*, damulin B

## Abstract

Alopecia, a prevalent yet challenging condition with limited FDA-approved treatments which is accompanied by notable side effects, necessitates the exploration of natural alternatives. This study elucidated the hair growth properties of *Gynostemma pentaphyllum* leaf hydrodistillate (GPHD) both in vitro and in vivo. Furthermore, damulin B, a major component of GPHD, demonstrated hair growth-promoting properties in vitro. Beyond its established anti-diabetic, anti-obesity, and anti-inflammatory attributes, GPHD exhibited hair growth induction in mice parallel to minoxidil. Moreover, it upregulated the expression of autocrine factors associated with hair growth, including VEGF, IGF-1, KGF, and HGF. Biochemical assays revealed that minoxidil, GPHD, and damulin B induced hair growth via the Wnt/β-catenin pathway through AKT signaling, aligning with in vivo experiments demonstrating improved expression of growth factors. These findings suggest that GPHD and damulin B contribute to the hair growth-inducing properties of dermal papilla cells through the AKT/β-catenin signaling pathway.

## 1. Introduction

Hair loss, or alopecia, is a condition which is likely to subconsciously impact one’s psychological health in the form of stress, low self-esteem, anxiety, and loss of confidence, which in turn can cause a domino effect and contribute to extensive hair loss. In addition to its social significance, hair loss can play a significant role in an individual’s self-esteem. Fifty percent of men will have been affected by hair loss by the age of 50, while forty percent of women will have experienced it by around age 70 [1,2,3]. Factors triggering hair loss can be either genetically inherited, environmentally influenced, or both. Androgenic alopecia, the predominant form of alopecia affecting both genders but exhibiting a greater impact on men due to elevated testosterone levels compared to women, may have a hereditary basis and can be worsened by additional environmental influences [4,5], while alopecia areata is caused by autoimmune destruction of hair follicles. Even though genetic predisposition plays a role in alopecia, extensive research has been conducted on how environmental factors such as smoking, diet, microbial inflammation, and UV radiation can cumulatively affect the emergence of this hair condition [6,7]. For instance, telogen effluvium and anagen effluvium can arise as a result of an unbalanced diet, stress, and chemotherapeutic agents, respectively, which suggests they are activated more by external factors rather than the genetic aspect [3,8]. In a nutshell, several internal disorders, hormonal and nutritional issues, and systemic intoxications, including hereditary characteristics, can have a significant role in the deterioration of hair texture, color, shine, elasticity, and manageability [9].

Some alopecia areata medications, such as baricitinib and ritlecitinib, have been newly approved by the FDA. Other established therapies approved by the Food and Drug Administration include options like low-level laser therapy (LLLT), topical or oral minoxidil, and oral finasteride [10,11,12]. In alopecia areata (AA), the inflammatory response is triggered by IFN-γ and IL-15 and mediated by the JAK1/2 signaling pathways in hair follicles. Both baricitinib and ritlecitinib have been demonstrated to be JAK inhibitors, acting as a therapeutic option for AA [11,12]. Furthermore, cyclosporine is frequently employed in alopecia areata treatment, either as a standalone therapy or in combination with other treatments. It is also utilized alongside certain antioxidants and immunosuppressants for managing androgenic alopecia. However, the mechanism of action of this drug is yet to be elucidated [13,14]. LLLT is a device-based therapy which uses low-intensity light to extend the anagen phase in hair follicles and enhance cellular proliferation in hair follicles, but its mechanism of action has yet not been fully elucidated [15]. Minoxidil, which was introduced as a drug for hypertension, was accompanied by hypertrichosis as a common side effect, which led to its development as a topical application to treat androgenic alopecia in men and women [16]. Although its mechanism of action is unclear, studies have speculated about its potential relevance, suggesting that its sulfated metabolite acts as a potassium channel opener, thereby producing nitric oxide (NO). NO is involved in various physiological processes, particularly vasodilation, and has been observed to increase capillary fenestrations, possibly linked to vascular endothelial growth factor (VEGF), which influences angiogenesis and cell functions such as survival and proliferation in dermal papilla cells [17,18]. Finasteride is a 5-alpha reductase inhibitor which converts testosterone into dihydrotestosterone, a powerful male hormone that causes hair loss [19]. However, considering the unpleasant side effects of these drugs, like erectile dysfunction, facial hypertrichosis, hypotension, skin irritation, headache, nasopharyngitis, upper respiratory tract infections, and many more, the emergence of safer alternatives is required and has already begun [10,12]. Out of necessity, studies have come up with some natural compounds with hair growth-inducing properties, with lesser to no aftereffects [20,21,22]. A similar strategy was followed in the present study, where we focused on the hair-inducing properties of *Gynostemma pentaphyllum*. *Gynostemma pentaphyllum (Thunb) Makino* is a perennial creeping plant from the Cucurbitaceae family. *G. pentaphyllum* has been used in Eastern medicine as a well-known culinary and medicinal plant [23]. The phytochemical examination of *G. pentaphyllum* unveiled its richness in saponin glycosides (gypsenosides), which were then investigated and found to have various pharmacological effects. Its apparent structural similarity to ginsenosides found in panax ginseng, already known for its numerous health benefits, including hair growth-promoting properties, led us to hypothesize that *G. pentaphyllum* might have similar characteristics [22,24,25]. *G. pentaphyllum* has been studied widely for its anti-diabetic, anti-obesity, and anti-anxiety effects [26,27], but not much is known about its hair growth-inducing properties. An intriguing aspect of this research study is that *G. pentaphyllum* is also utilized as a sweetener [28], and several studies have examined the influence of olfactory receptors on hair growth. It has been observed that human hair follicles possess chemosensory capabilities, with the specific activation of OR2AT4 being essential for sustaining hair follicle growth [29]. While additional research may be necessary to reinforce this hypothesis, it is noteworthy to mention its potential significance.

Dermal papilla cells are considered a central node in hair growth as they are responsible for regulating the hair growth cycle [30,31]. In the present study, we focused on the proliferative influence of *G. pentaphyllum* leaf hydrodistillate (GPHD) on human dermal papilla cells (hDPCs) in vitro as well as in vivo, with particular attention to the method of extraction to ensure complete purity and address any safety concerns. Furthermore, we analyzed the mechanism through which *G. pentaphyllum* and its key component damulin B could exhibit hair-inducing properties by activating the Wnt/β-catenin pathway via the AKT/ GSK3β pathway, making them potential candidates for the prevention of alopecia.

## 2. Materials and Methods

### 2.1. Cell Culture, Reagents, and Antibodies

An immortalized human dermal papilla cell (hDPC) line retaining the characteristics of primary DPCs was used for the current study [32,33]. This cell line expresses alkaline phosphatase, versican, alpha-smooth muscle actin, and biglycan. Human dermal papilla cells were maintained in Dulbecco’s modified eagle medium (DMEM) (Gibco, Grand Island, NY, USA) supplemented with 10% heat inactivated fetal bovine serum (Corning, NY, USA) in humidified air containing 5% CO_2_ at 37 °C. Antibodies against GAPDH and PCNA were purchased from Santa Cruz Biotechnology (Santa Cruz, CA, USA). Antibodies against β-catenin, GSK3β, P-GSK3β (Ser9), AKT, and P-AKT(Ser473) were purchased from Cell Signaling Technology (Danvers, MA, USA). Damulin B was obtained from ChemFaces Biochemical (Wuhan, China). Commercially available 5% minoxidil was used for in vivo experiments.

### 2.2. Formulation of Cell Culture Media Using G. pentaphyllum Leaves Extract

The hydrodistillate of *G. pentaphyllum* was generated by Chunjieh Cooperation (Jeju City, Korea) as previously described [34]. Around 20 kg of dried *G. pentaphyllum* leaves was mixed with 1.2 tons of distilled water and heated for 15 h at 109 °C under a pressure of 1.5 atmospheres. The vapor produced during this process was condensed using a closed-loop system which subsequently underwent cooling. This hydrodistillate was heated again at 95 °C for 1 h to be sterilized. This hydrodistillate was considered 100% (*v/v*) and was directly utilized for in vivo experiments. A cell culture medium, hereafter referred to as the GPHD culture medium, was formulated for in vitro applications. The GPHD culture medium was prepared by substituting distilled water with 100% hydrodistillate when adding DMEM powder, as opposed to the typical practice of using distilled water in the formulation of regular DMEM media. The pH was adjusted to 7.2 using bicarbonate. This medium was filter sterilized, and 10% fetal bovine and 1% antibiotics were added. This GPHD culture medium was then mixed with regular DMEM and the desired percentage of GPHD (*v*/*v*) was prepared. 

### 2.3. Trypan Blue Cell Counting Assay

hDPCs were treated with selected concentrations of minoxidil, GPHD, and damulin B and incubated for 24 h. The collected suspension was mixed with trypan blue 1:1 and viable cells were counted using a hemocytometer. 

### 2.4. Animal Experiments

Three-week-old male C57BL/6 mice were purchased from Japan SLC (Shizuoka, Japan) and allowed to adapt for 1 week at 21 ± 2 °C and 50% ± 5% relative humidity under a 12 h light/12 h dark cycle before use in experiments. The experimental mice were randomly allocated into 3 groups, namely control, GPHD, and minoxidil, each group consisting of 5 mice. Minoxidil was used as a positive control. The control group was fed a normal diet and distilled water (~0.9 mL/day) for 28 d. The GPHD group was fed a normal diet and GPHD (~0.9 mL/day) for the same duration, whereas the minoxidil group was fed a normal diet and distilled water (~0.9 mL/day) for 28 d prior to shaving. On day 28, the mice were anesthetized with zoletyl and the dorsal fur of the mice was shaved using an animal clipper. For an additional 15 days, the control group continued with a normal diet and distilled water (~0.9 mL/day) while the GPHD group received a normal diet and GPHD (~0.9 mL/day). The minoxidil group was fed similarly to control group but minoxidil was topically applied on the shaved dorsal region every day for the next 15 days. As previously stated, determining the concentration of GPHD is challenging compared to determining that of minoxidil due to its nature as an extract, and applying it in an ointment form would further dilute the concentration due to viscosity. Even if applied topically, we anticipated that its effect would not have matched that of minoxidil, requiring more frequent applications, which would have disrupted methodological uniformity. Moreover, since the extract showed no toxicity when administered orally, priority was given to oral administration to ascertain the volume of GPHD at least. For visual analysis of improvement in hair growth, images were taken on days 0, 7, 11, 13, and 15. The animal protocol was approved by the Institutional Animal Care and Use Committee of Kyung Hee University (KHSASP-21-309).

### 2.5. Hair Growth Score Analysis of Improvement in Hair Growth

A hair growth score was calculated according to a previous report [35] in the following way. The hair growth phase was partitioned into three distinct areas: a pink region denoted as A, representing minimal hair growth; a grey section marked as B, indicating slight hair growth; and a black area labeled as C, signifying nearly complete hair growth. Hair growth score= (Area of A × 0) + (Area of B × 1) + (Area of C × 2)/total area. Areas A, B, and C of mouse dorsal skin were converted into numbers using the Image J program. 

### 2.6. Histopathological Analysis

For histopathological analysis, mouse dorsal skins were immediately dissected out, fixed in 10% neutral buffered formalin solution, embedded in paraffin, and cut at a thickness of 5 µm. Each section was subjected to H&E (BBC Biochemical, Mount Vernon, WA, USA) staining. Hair cycles were assessed and categorized based on quantitative histomorphometry as described [36] using images obtained from hematoxylin and eosin (H&E) staining.

### 2.7. Real-Time PCR for Mouse Skin Tissue and hDPCs

To analyze the expression level of various growth factors in mouse dorsal skin tissue, RNA was extracted from 100 µg of each mouse’s skin with TRIzol™ Plus RNA Purification Kit (Invitrogen, Carlsbad, CA, USA). Similarly, total RNA was extracted from hDPCs using TRIZol reagent (Invitrogen, Carlsbad, CA, USA), and cDNA was reverse-transcribed from 1 µg of total RNA with the RvertAidTM First Strand cDNA synthesis Kit (Fermentas, Glen Burnie, MD, USA) according to the manufacturer’s instructions. Real-time PCR was carried out using Power SYBR^®^ Green PCR Master Mix (Applied Biosystems, Foster City, CA, USA) using the QuantStudio 5 system (Thermo Fisher Scientific, Waltham, MA, USA). The thermal cycling program for all target and reference genes for mouse skin tissue was as follows: pre-denaturation (2 min at 50 °C), denaturation (10 min at 95 °C), annealing, and extension (15 s at 95 °C, 30 s at 60 °C, 30 s at 72 °C) for 40 cycles. The melting curve analysis condition was as follows: 15 s at 95 °C, 1 min at 60 °C, and 15 s at 95 °C. GAPDH was used to normalize the expression of each gene, which was presented using the 2^−ΔΔCt^ method. The primers are listed as follow: VEGF forward (TGGTGGACATCTTCCAGGAG); VEGF reverse (GGAAGCTCATCTCTCCTATGTG); KGF forward (CGCAAATGGATACTGACACG); KGF reverse (GGGCTGGAACAGTTCACACT); IGF-1 forward (ACTGGAGATGTACTGTGCCC); IGF-1 reverse (GATAGGGACGGGGACTTCTG); HGF forward (CATTGGTAAAGGAGGCAGCTATAAA); HGF reverse (GGATTTCGACAGTAGTTTTCCTGTAGG); GAPDH forward (ACCACAGTCCATGCCATCAC); GAPDH reverse (TCCACCACCCTGTTGCTGT). Given that the cell line utilized in the in vitro culture is of human origin, the primers employed for gene identification are specified as follows: VEGF forward (TCTTCAAGCCATCCTGTGTG) and reverse (GCGAGTCTGTGTTTTTGCAG); KGF forward (ACTCCAGAGCAAATGGCTAC) and reverse (CCACTGTCCTGATTTCCATG); IGF-1 forward (TCAACAAGCCCACAGGGTAT) and reverse (CGTGCAGAGCAAAGGAT); HGF forward (TGTGGGTGACCAAACTCCTG) and reverse (AGCGTACCTCTGGATTGCTT); and GAPDH forward (TGCATCCTGCACCACCAACT) and reverse (TGCCTGCTTCACCACCTT).

## 3. Results

### 3.1. GPHD Promotes Hair Growth in Mice

Throughout the current experiment, we used *G. Pentaphyllum* hydrodistillate (GPHD), which has been reported to be safe and contains many active ingredients [34]. To examine the hair growth-inducing effects of GPHD on mice, random groups consisting of five mice in each group were administered the respective compounds. The schedule for the in vivo experiments is summarized in Figure 1A. The control group was fed a normal diet with water (~0.9 mL/day), the GPHD group was fed a normal diet with the equivalent amount of GPHD (~0.9 mL/day), while in the minoxidil group, minoxidil was topically applied on the depilated dorsal regions of these mice. Minoxidil was chosen over finasteride due to its potential as a therapeutic drug for promoting hair growth, while finasteride serves as a protective barrier against hair loss induced by hormonal factors. Notably, topical minoxidil was chosen over its oral counterpart, considering its perceived safer profile and the potential systemic side effects associated with oral administration [37]. Moreover, the complexity of applying oral minoxidil in an experimental setting given its tablet formulation influenced the decision in favor of the topical form. Figure 1A displays the progression of hair growth through images taken at 7, 11, 13, and 15 days after shaving. The GPHD group exhibited superior hair growth compared to the control group, approaching levels observed in the minoxidil group. Figure 1B illustrates one of the images used for quantifying hair growth on the dorsal skin of mice, with distinct colors denoting various phases of hair growth. A hair growth score was calculated according to the method described in a previous report [35], and pink, grey, and black areas were quantified using Image J software (v1.54i) to determine the hair growth score, revealing a 2.8–2.9-fold increase in hair growth for both the minoxidil and GPHD groups. Skin tissue was collected on day 15 after shaving and was processed using H&E staining for histological analysis. Figure 1C shows the H&E staining of longitudinal and transverse sections of hair follicles in all three groups. The GPHD and minoxidil groups exhibited mature hair follicles in the dermis region which were absent in the control group. Quantitative histomorphometry, following the method outlined by S. Müller-Röver et al. [36], revealed that both GPHD and minoxidil facilitated the transition from telogen to anagen, indicating that GPHD possesses growth-promoting properties similar to minoxidil. Additionally, Figure 1D shows the mRNA expression of growth factors (VEGF, IGF-1, HGF, and KGF) in dorsal skin tissue samples. Minoxidil demonstrated a 3–4-fold induction of the expression of growth factors, while GPHD exhibited a comparatively higher expression than the control, though not as high as minoxidil.

### 3.2. Minoxidil, GPHD, and Damulin B Enhance the Proliferation of Human Dermal Papilla Cells (hDPCs)

To explore the therapeutic effects of *G. Pentaphyllum* in vitro, the culture medium containing GPHD referred to in Materials and Methods was made. We also examined the effects of damulin B, a hydroxylated dammarane-type glycoside, as the analysis of GPHD via ultra-high performance liquid chromatography–quadrupole/time-of-flight mass spectrometry (UHPLC-Q/TOF MS) had previously shown it to be a predominant component [34]. Human dermal papilla cells (hDPCs) were treated with the indicated concentration of minoxidil (Figure 2A), GPHD (Figure 2B), and damulin B (Figure 2C,D) for 24 h. A trypan blue cell counting assay revealed the cell-proliferative effects of each reagent, while no toxicity was observed (Figure 2E).

### 3.3. GPHD, Damulin B, and Minoxidil Enhance the Expression of Growth Factors in hDPCs

Based on the outcome observed in the in vivo experiment, we postulated that the stimulation of hair growth could be attributed to the presence of growth factors, as depicted in Figure 1D. Subsequently, we opted to investigate whether analogous results could be replicated in an in vitro environment. Many growth factors, including VEGF, IGF, KGF, and HGF, have been reported to stimulate hair growth via various signaling pathways [38]. Real-time PCR revealed that minoxidil, GPHD, and damulin B significantly induced the mRNA expression of the indicated growth factors in hDPCs in a dose-dependent manner (Figure 3A–C).

### 3.4. Activation of Wnt/β-Catenin and Akt Signaling Pathways by Minoxidil, GPHD, and Damulin B

Studies have suggested a focal role of the Wnt/β-catenin signaling pathway in follicle cycling and hair morphogenesis in embryonic and adult life [38,39]. Additionally, studies have suggested that minoxidil activates the β-catenin pathway in dermal papilla cells to prolongate the anagen phase in the hair follicle cycle [40]. It is widely acknowledged that β-catenin induces the transcription of genes associated with cellular proliferation, encompassing diverse growth factors; hence, its deficiency within dermal papilla cells can disrupt the niche, hindering hair growth [41]. Thus, we focused our study on the Wnt/β-catenin signaling pathway as a plausible mechanism of hair growth-promoting activity. Minoxidil, GPHD, and damulin B induced the expression of β-catenin in a concentration- (Figure 4A–C) and a time-dependent manner (Figure 4D,E) in hDPCs. GSK3β in its active, non-phosphorylated state is responsible for the ubiquitination of β-catenin. However, an elevation in the expression of phosphorylated GSK3β-Ser9, indicating its inactive form, was observed, suggesting the stabilization of β-catenin. Several studies have emphasized the significance of phosphoinositide 3-kinase and the AKT (PI3K/AKT) pathway in the maintenance, proliferation, and even de novo synthesis of hair follicle regeneration [20,42,43]. Our results demonstrated that the level of phosphorylated AKT, which mediates the signals of various growth factors, was also significantly increased by these reagents [44]. Notably, significantly elevated expression of the proliferation marker proliferating cell nuclear antigen (PCNA) was observed in the presence of minoxidil, with a modest increase noted in the presence of both GPHD and damulin B, indicating the induction of DNA synthesis. Collectively, these results suggest that minoxidil, GPHD, and damulin B induce cell proliferation through similar mechanisms of activating growth factors and the β-catenin signaling pathway (Figure 4).

### 3.5. The Wnt/β-Catenin Pathway Is Activated via AKT in the Presence of GPHD and Damulin B

Next, we investigated whether the Akt signaling pathway is directly involved in the induction of β-catenin. Results from the time-dependent experiment indicated a notable increase in the expression of P-AKT and P-GSK3β during the initial exposure (1.5 h) to GPHD and damulin B. In contrast, the expression of β-catenin was observed at a later time point (24 h), as depicted in Figure 4D,E. Based on these temporal patterns, we suspected that Akt may act as an upstream of β-catenin. GPHD- and damulin B-induced Akt activation, the phosphorylation of GSK-3β, and induction of β-catenin were significantly blocked when hDPCs were pretreated with wortmannin, a specific inhibitor of PI3-kinase (Figure 5A,B). However, the expression level of PCNA was hardly affected by wortmannin, indicating that PCNA induction is not directly regulated via PI3-kinase and the Akt signaling pathway. The expression level of growth factors was also significantly suppressed by wortmannin under the same condition (Figure 5C,D). These results indicate that the following signaling cascade is activated in hDPCs: GPHD and damulin B activate the PI3-kinase/Akt pathway, which induces β-catenin, which in turn activates the transcription of various growth factors.

## 4. Discussion

The hair cycle undergoes progression and retrogression as it goes through certain phases, named the anagen, catagen, and telogen phase, anagen being the longest phase during which the cells are actively growing, while catagen and telogen being the apoptosis and resting phases, respectively [30,45]. The hair follicle contains diverse cell types which are essential for its growth and structure. Hair matrix cells, situated close to the dermal papilla, interact with it to stimulate the formation of epithelial elements like the inner root sheath and medulla. The hair shaft, primarily composed of keratinocytes, and the inner and outer sheaths collectively constitute the epidermal components of the hair follicle. The dermal papilla supplies nutrients for hair growth and follicle maintenance, forming a multicellular tissue structure crucial for inducing hair growth [46,47]. Dermal papilla cells are eminently active during the anagen phase, owing to their sophisticated communication within the hair follicle niche, providing certain instructive growth factors and cytokines for regulation of the hair shaft [31]. This makes the dermal papilla notably crucial, as its population and activity markedly modulate the prolongation or abruption of these phases, which ultimately determines the size of the hair shaft [48]. Human hair follicle growth and morphogenesis are primarily regulated by the cell signaling pathways Wnt, Shh, Notch, and BMP. In the induction phase of hair follicle formation, the Wnt pathway functions as a master regulator, including its upstream regulators like AKT and GSK3β [49,50]. In addition, it has been shown that the development and regeneration of hair follicles are significantly influenced by Wnt/-β-catenin signaling to the extent that β-catenin deletion in the embryonic epidermis abrogates hair follicle morphogenesis [51].

The genus *Gynostemma* has been extensively explored in traditional medicine for its various properties in treatment of diabetes, hypertension, obesity, and hepatosteatosis. However, limited knowledge exists regarding its potential to stimulate hair growth [26,27]. In the current study, we have successfully showcased the ability of *G. pentaphyllum* extract to enhance the hair growth-inducing properties of dermal papilla cells. A notable study has claimed that *G. pentaphyllum* extract elicited the induction of nitric oxide and evidently promoted vasodilation in bovine aorta endothelial cells [52], which coincidentally aligns with the mechanism of action of minoxidil. The study aimed to investigate the potential of *G. pentaphyllum* in promoting hair growth-inducing properties in dermal papilla cells [16]. Additionally, another study suggested that minoxidil activates the β-catenin pathway, which prolongs the anagen phase [40]. The Wnt/β-catenin signaling pathway, along with its target genes, is recognized as a crucial regulator of dermal papilla cell proliferation and the maintenance of the hair follicle [50,53]. In the Wnt/β-catenin canonical pathway, Wnt binds to the frizzled receptor and the low-density lipoprotein-related protein (LRP), and this complex inactivates GSK3β by phosphorylation, which is generally responsible for cytoplasmic ubiquitin-mediated degradation of β-catenin, thus allowing its translocation to the nucleus to subsequently trigger the activation of downstream targets which regulate the proliferative properties of the cells [38,39,54]. Research has revealed that the activation of the AKT pathway, as well as its downstream regulators such as β-catenin and GSK3β, extensively participates in various cellular signaling pathways, encompassing cell survival, angiogenesis, migration, and metabolism [55]. The data we obtained demonstrated that *G. pentaphyllum* extract and damulin B alone have the ability to activate the AKT pathway. Consequently, this activation leads to the phosphorylation and subsequent inactivation of downstream GSK3β, preventing it from phosphorylating β-catenin. Subsequently, this facilitates the nuclear translocation of β-catenin, leading to the upregulation of its target genes, including growth factors and various cytokines. These growth factors and cytokines exert an autocrine influence on the proliferation and differentiation of dermal papilla cells, as observed in our study. Secretion of IGF-1 from dermal papilla cells has been reported to stimulate hair follicle growth via the PI-3 kinase pathway [56]. The expression of VEGF was also found to be crucial for hair growth as it helps in the direct proliferation of dermal papilla cells and/or stimulation of local vascularization [57]. HGF secreted by DPCs stimulates the growth of keratinocytes, highlighting the significance of epithelial–mesenchymal interactions in hair follicle maintenance and induction [58]. KGF has also been recognized for its ability to support the survival of hair follicles and provide protection against detrimental external factors [59]. Our data indicated *G. pentaphyllum* extract and its potent component damulin B increased the mRNA levels of VEGF, IGF-1, HGF, and KGF (Figure 1D and Figure 3B,C). To inspect the relationship between PI3K/AKT signaling and its downstream targets with GPHD and damulin B, we employed a PI3K inhibitor, wortmannin, observing its effects on potential downstream pathways (Figure 5). Our findings indicated that wortmannin effectively inhibited AKT phosphorylation, which remained suppressed even in the presence of GPHD and damulin B, suggesting that these compounds activate the β-catenin pathway via AKT signaling. Furthermore, in the presence of wortmannin, the expression of growth factors markedly decreased and remained low, even upon the introduction of GPHD and damulin B, hinting at the possibility that AKT activation may be triggered by the autocrine effect of growth factors, subsequently enhancing proliferation. The expression levels of β-catenin, P-AKT, and P-GSK3β were significantly decreased in the presence of wortmannin, which suggests *that G. pentaphyllum* could induce the expression of growth factors via the AKT/GSK3β pathway. In conclusion, our data suggest that the autocrine signaling pathway is amplified by *G. pentaphyllum* extract and damulin B; the expression of growth factors is induced, and these growth factors are secreted to bind to the cell receptors and activate the signaling pathway to induce the expression of growth factors. These results are summarized in Figure 6. Currently, the exact molecular targets of GP extract and damulin B are unknown.

Hair loss can be perceived as a flaw by the observer and sufferer due to the norms of today’s beauty standards. Studies have shown that extent of hair loss can be proportional to psychological and/or physical stress [60] and may increase day by day. Androgenic alopecia represents a prevalent type of hair loss, impacting approximately 85% of men and 40% of women [61]. Androgenic alopecia (AGA), commonly associated with elevated dihydrotestosterone (DHT) production, stands as the main cause of hair loss in men. The enzymatic activity of 5-α reductase, responsible for the conversion of testosterone to dihydrotestosterone (DHT), is elevated in the scalp affected by balding. Consequently, increasing DHT levels in the balding scalp, accompanied by an augmentation in the number of DHT receptors on the hair follicles in that region, eventually lead to the miniaturization of hair follicles [62,63]. Finasteride, as the sole FDA-approved drug currently available, inhibits 5-α reductase to diminish the production of dihydrotestosterone (DHT), emphasizing the importance of investigating potential interventions aimed at mitigating androgenic alopecia (AGA) by targeting DHT. Our future studies include assessing the efficacy of GPHD and damulin B against the detrimental effects of DHT. 

## 5. Conclusions

In conclusion, based on our current knowledge, as we seek a natural, low-risk alternative to the current therapeutic options for hair loss, we propose *G. Pentaphyllum* extract as a promising candidate for use in addressing alopecia.

## Figures and Tables

**Figure 1 nutrients-16-00985-f001:**
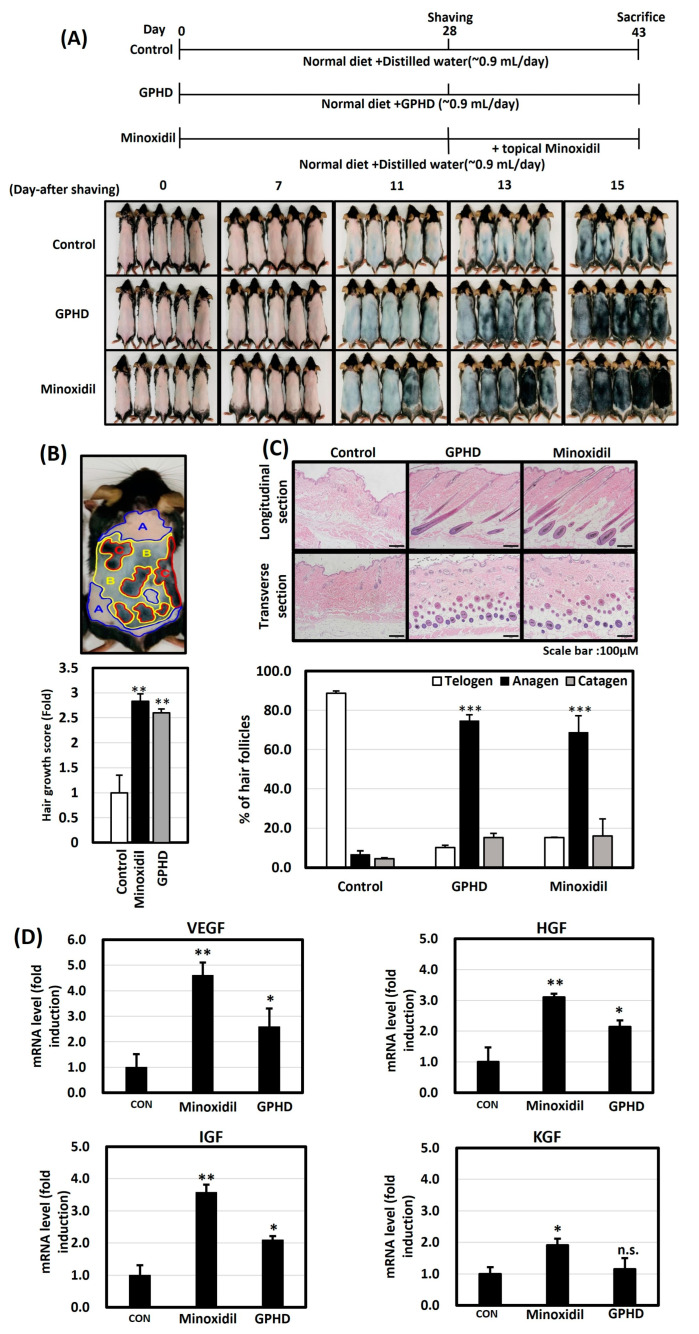
GPHD promotes hair growth in mice. (**A**) Six-week-old mice were fed a normal diet and distilled water (~0.9 mL/day) or GPHD, while the minoxidil group was fed a normal diet and distilled water (~0.9 mL/day) with an additional topical application of minoxidil for 15 days before being sacrificed. Images show hair growth progression on mouse dorsal skin on day 0, 7, 11, 13, and 15 after depilation. (**B**) Hair growth score was calculated in the following way: Hair growth score = (Area of A × 0) + (Area of B × 1) + (Area of C × 2)/total area. (**C**) H&E staining of mouse dorsal skin sections showed morphological changes in hair follicles and quantitative histomorphometry showed changes in hair cycle phases following treatment with minoxidil and GPHD. (**D**) Growth factors in each mouse group’s dorsal skin tissue were analyzed by real-time PCR. The results shown are representative of three independent experiments. *p*-value < 0.05 was considered statistically significant; individual *p*-values (* *p* < 0.05; ** *p* < 0.005; *** *p* < 0.0005) are indicated. n.s; not significant.

**Figure 2 nutrients-16-00985-f002:**
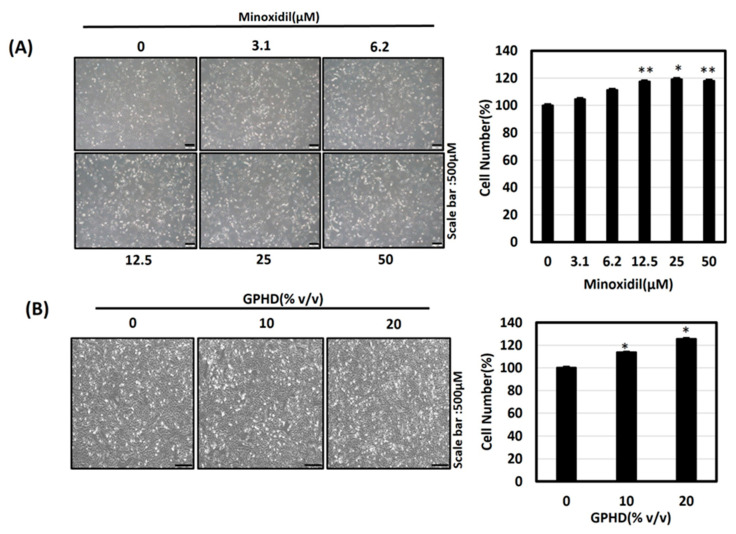
Minoxidil, GPHD, and damulin B enhance the proliferation of human dermal papilla cells (hDPCs). hDPCs were treated with the indicated concentration of minoxidil (**A**), GPHD (**B**), and damulin B (**D**) for 24 h, and cell numbers were measured. Microscopic images of hPDCs were taken and cell numbers were examined via trypan blue cell counting assay. (**C**) Chemical structure of damulin B. (**E**) MTT assays were performed under identical conditions. The results shown are representative of three independent experiments. *p*-value < 0.05 was considered statistically significant; individual *p*-values (* *p* < 0.05; ** *p* < 0.005) are indicated.

**Figure 3 nutrients-16-00985-f003:**
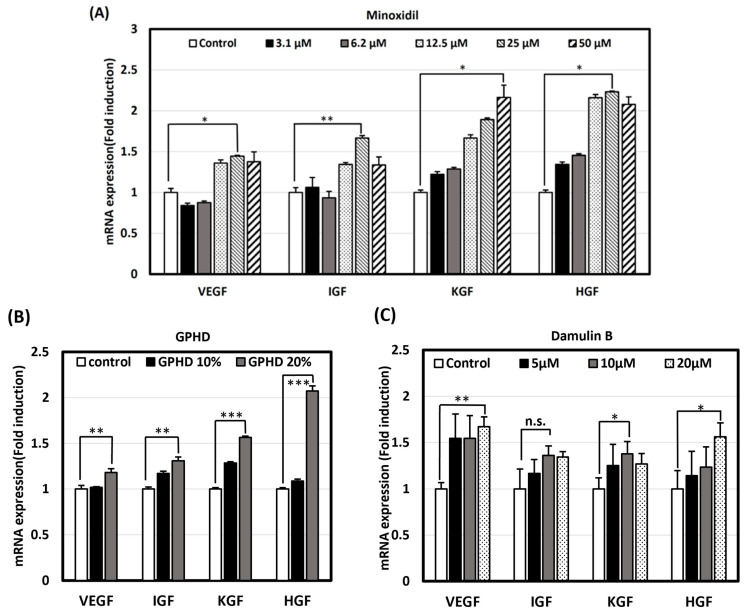
Minoxidil, GPHD, and damulin B induce the expression of growth factors in hDPCs. hDPCs were treated with minoxidil (**A**), GPHD (**B**), and damulin B (**C**) for 24 h in a concentration-dependent manner. Then, real-time PCR was performed to measure the mRNA level of each growth factor. The results shown are representative of three independent experiments. *p*-value < 0.05 was considered statistically significant; individual *p*-values (* *p* < 0.05; ** *p* < 0.005; *** *p* < 0.0005) are indicated. n.s; not significant.

**Figure 4 nutrients-16-00985-f004:**
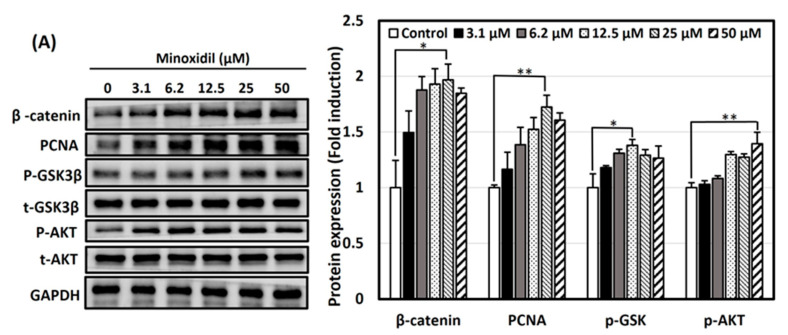
Minoxidil, GPHD, and damulin B activate the Wnt/β–catenin pathway and Akt signaling pathway in hDPCs. hDPCs were treated with minoxidil, GPHD, and damulin B in a concentration-dependent (**A**–**C**) or time-dependent manner (**D**,**E**). Western blot analysis was performed, and the level of the indicated protein was quantified. The results shown are representative of three independent experiments. *p*-value < 0.05 was considered statistically significant; individual *p*-values (* *p* < 0.05; ** *p* < 0.005; *** *p* < 0.0005) are indicated.

**Figure 5 nutrients-16-00985-f005:**
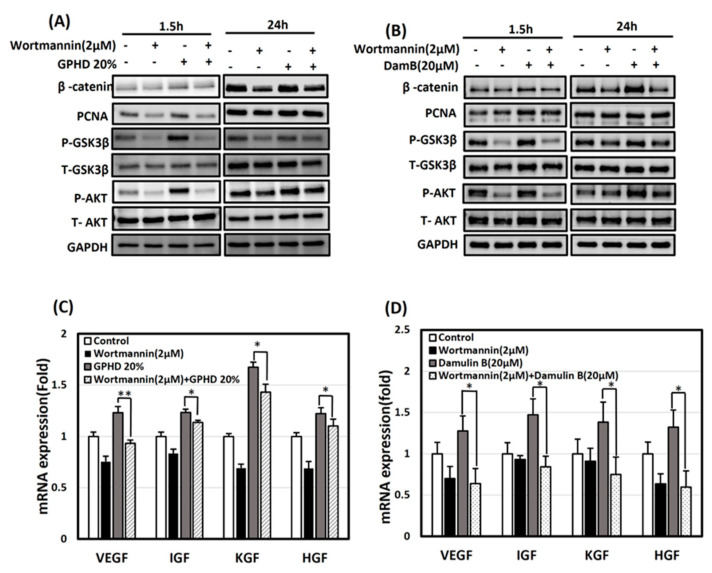
Activation of the Wnt/β –catenin pathway via AKT in the presence of GPHD and damulin B. hDPCs were treated with GPHD/damulin B in the presence of and/or absence of wortmannin. Western blot analysis (**A**,**B**) and real-time PCR analysis (**C**,**D**) were performed. The results shown are representative of three independent experiments. *p*-value < 0.05 was considered statistically significant; individual *p*-values (* *p* < 0.05; ** *p* < 0.005) are indicated.

**Figure 6 nutrients-16-00985-f006:**
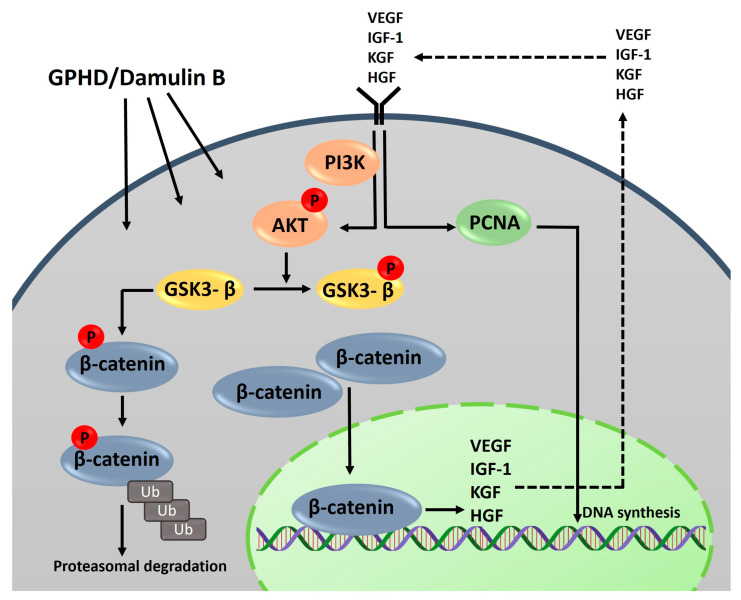
Our results are summarized in this diagram. GPHD and damulin B activate the PI3K and Akt pathways, which leads to the induction of several growth factors. We assume that these growth factors are secreted to the medium and then act as growth factors for hDPCs in an autocrine manner. As a result, this signal loop might be amplified by GPHD and damulin B. The dashed line indicates the speculation of the authors.

## Data Availability

Data are contained within the article.

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
