# Peer review of "Gynostemma pentaphyllum Hydrodistillate and Its Major Component Damulin B Promote Hair Growth-Inducing Properties In Vivo and In Vitro via the Wnt/β-Catenin Pathway in Dermal Papilla Cells"

_nutrients, 2024, doi:10.3390/nu16070985_

Round 1

Reviewer 1 Report

Comments and Suggestions for Authors

In this study, Kovale et al. study the impact of a natural extract on hair growth in vivo and on dermal papilla cells in vitro. While this natural compound could be interesting for treating some hair diseases, I have some comments about the present study:

1) elevated testosterone levels compared to women, may have a hereditary basis. Please provide corresponding references for this claim.

2) In the introduction, the authors cite Minoxidil and finasteride as treatment for alopecia. What about the use of cyclosporine?

3) DP cells are responsible for regulating the hair growth cycle.: yes, but not the only ones. Also, lease add corresponding references for this claim.

4) Many natural extracts are also known to activate either taste or olfactory receptors which are now become important in regulating hair follicle physiology, it might be interesting to mention at least this in the introduction given that G. pentaphyllum has been used as sweetener.

5) in humidified air containing CO2: What % of CO2? 5%

6) Where the Human dermal papilla cells primary cells extracted from human scalp hair follicle or purchased from any company?

7) Why minoxidil was used as positive control and not Finasteride for example? A justification of the use of this positive control would be important. 

8) Why GPHD was used orally and not topically as GPHD? The different modes of application may affect the outcome of the results and make it difficult to compare. How the authors can be sure that GPHD has no side effects when orally administered?

9) The use of Hair growth score is good but since the authors have also some H&E sections, it would be great to show also quantitative hair cycle staging (see Müller-Röver S, J Invest Dermatol. 2001). The hair growth score gives an idea of the stage (anagen, catagen ,telogen) but not a precise staging (Anagen I, II, III, ...).

10) Experiments using human dermal papilla fibroblasts shouldn't be perform in 2D but in 3D spheroids, since these cells are known to lose their inductivity properties after a couple of passage after isolation and can be used for a few additional passage when culture in 3D spheroids. It would then be more useful and translationally relevant to repeat the experiment with DP spheroids. Also, did the authors check that these hDP fibroblasts still have their inductivity properties using for example alkaline phosphatase and/or versican? If not, how can the authors be sure that the cells were still DP fibroblasts and not "normal" fibroblasts?

11) Looking at the expression of the growth factors is great, however these growth factors are mainly expressed in the hair matrix and outer root sheath of hair follicle. Why not having look at the expression of these growth factors in vivo to show which pathway(s) is/are modulated by GPHD?

12) In the Figure 3B, what is GPE? Is it GPHD?

13) In the summary diagram, the authors assumed that the growth factors are secreted to the medium and act as growth factors for hDPC. This claim is too strong based on the study results, especially the fact that these growth factors act via an autocrine loop. To make this claim, the authors would need at least to perform experiments with neutralizing antibodies. 

14) In the discussion the authors say that the expression of IGF-1 has been reported to stimulate the hair elongation and promote the anagen phase in of dermal papilla cells [44]. The reference cited here does not show this claim and IGF-1 is the most potent anagen prolonging growth factor in hair follicle but not in dermal papilla. Please revised this sentence in the discussion.

15) as they are a known an instructive niche for progenitor and epithelial stem cells in their development and Regeneration [35]: Apart the fact that this sentence is absolutely similar to the title of the reference 35, Isn't the source of the hair follicle epithelium located mainly in the bulge?

Comments on the Quality of English Language

Some grammar mistakes need to be corrected. See a few examples below:

1) therapies drugs : it should be either drug therapies or therapeutical drugs.

2) an emergence of safer alternatives is rising: the emergence of safer alternative is required and start to rise

3) Similar strategy was followed in present study: Similar strategy was followed in the present study

4) anti-obesity, and anti-anxiety effects, not much is known about: anti-obesity, and anti-anxiety effects [21, 22], but not much is known about

5) as they are a known an instructive niche: as they are a known instructive niche

6) the hair elongation and promote the anagen phase in of dermal papilla cells: the hair elongation and promote the anagen phase of hair follicles

Author Response

Reviewer 1:

In this study, Kovale et al. study the impact of a natural extract on hair growth in vivo and on dermal papilla cells in vitro. While this natural compound could be interesting for treating some hair diseases, I have some comments about the present study:

I would like to sincerely thank you for reading my manuscript very carefully and giving me a lot of good advice and points. I expect that the quality of my manuscript will improve dramatically due to these points. I have answered the issues you have pointed out to the best of my ability, and I hope that these answers will be satisfactory.

1) elevated testosterone levels compared to women, may have a hereditary basis. Please provide corresponding references for this claim.

Reply: The following reference (reference number 5) was added at line 39.

Richards, J.B., et al., Male-pattern baldness susceptibility locus at 20p11. Nat Genet, 2008. 40(11): p. 1282-4.

2) In the introduction, the authors cite Minoxidil and finasteride as treatment for alopecia. What about the use of cyclosporine?

Reply: Many studies were conducted to test whether cyclosporine could be developed as a treatment for alopecia areata, which is caused by an autoimmune response. However, it was not approved by the FDA. Instead, we've added descriptions of two recently FDA-approved treatments, Baricitinib and Ritlecitinib, for alopecia areata on lines 48-53 along with two new references (ref 11~12).

3) DP cells are responsible for regulating the hair growth cycle.: yes, but not the only ones. Also, lease add corresponding references for this claim.

Reply: Two new references(28, 29) describing the role of DP cells on induction and regeneration of hair follicles are added at line 89.

4) Many natural extracts are also known to activate either taste or olfactory receptors which are now become important in regulating hair follicle physiology, it might be interesting to mention at least this in the introduction given that G. pentaphyllum has been used as sweetener.

Reply: The suggested possibilities are newly described along with two new references (26, 27) at line 82~87.

5) in humidified air containing CO2: What % of CO2? 5%

Reply: 5% was added at line 103.

6) Where the Human dermal papilla cells primary cells extracted from human scalp hair follicle or purchased from any company?

Reply: We used the immortalized human dermal papilla cells retaining the characteristics of the primary DPC. This cell line is reported to express DPC markers such as alpha-smooth muscle actin and biglycan, and also respond to Wnt-beta-catenin and BMP signal pathways, which are required for hair-inducing activity of DPCs. Descriptions of cell line we used in this study are newly included at line 99 along with the new reference (30) showing its properties. 

7) Why minoxidil was used as positive control and not Finasteride for example? A justification of the use of this positive control would be important. 

Reply: This study is primarily focused on exploring the properties of GPHD to promote hair growth rather than prevent hormone-related damage. This is why we chose minoxidil over finasteride, which acts as a shield against hormone-induced hair loss. We added this explanation on line 194. The differences between the underlying mechanisms of minoxidil and finasteride were discussed in detail in the Introduction.

8) Why GPHD was used orally and not topically as GPHD? The different modes of application may affect the outcome of the results and make it difficult to compare. How the authors can be sure that GPHD has no side effects when orally administered?

Reply: I absolutely agree with this point and put a lot of thought into the administration method when I first designed our experiment. Since GPHD is essentially a liquid, it is very difficult to develop a formulation that can be applied topically. It is also very difficult to compare the concentration of GPDH to that of minoxidil. For these reasons, we decided that oral administration would be more effective than topical administration.

As mentioned above, GPHD is a distillate of water, so it is expected to be very safe. In our previous experiment, we showed that GPHD reduces cisplatin-induced kidney toxicity. We fed GPHD to mice for 32 days and saw no toxicity in kidney damage markers or histology. We also found no toxicity in HEK293 cells. These results can be found in reference 31GPHD did not show any toxicity in hDPC in current study (figure 2E).

9) The use of Hair growth score is good but since the authors have also some H&E sections, it would be great to show also quantitative hair cycle staging (see Müller-Röver S, J Invest Dermatol. 2001). The hair growth score gives an idea of the stage (anagen, catagen, telogen) but not a precise staging (Anagen I, II, III, ...).

Reply: Thank you for introducing a very meaningful methodology to analyze hair cycle phases. Indeed, the current study was aimed at identifying the hair growth effects of GPHD and the underlying basic mechanisms. As a follow-up study, we plan to further investigate the effects of GPHD on the hair cycle stages, the more detailed mechanisms involved, and the possibility of developing it as a drug. We believe that utilizing the methodology you have introduced will allow us to obtain more detailed and meaningful results, and we hope that you will allow us to use the results of this analysis in our follow-up studies.

10) Experiments using human dermal papilla fibroblasts shouldn't be perform in 2D but in 3D spheroids, since these cells are known to lose their inductivity properties after a couple of passage after isolation and can be used for a few additional passages when culture in 3D spheroids. It would then be more useful and translationally relevant to repeat the experiment with DP spheroids. Also, did the authors check that these hDP fibroblasts still have their inductivity properties using for example alkaline phosphatase and/or versican? If not, how can the authors be sure that the cells were still DP fibroblasts and not "normal" fibroblasts?

Reply: Thank you for bringing up a good point. We realize that it is important to use the default cells of the 3D spheroid to get more accurate results, but it is also true that it is too costly and time consuming to generate a large amount of data. To overcome these issues, we used the immortalized human dermal papilla cells that express DPC markers such as alpha-smooth muscle actin and biglycan, and also respond to Wnt-β-catenin and BMP signal pathways required for hair-inducing activity of DPCs. Please refer to the reference 30.

11) Looking at the expression of the growth factors is great; however these growth factors are mainly expressed in the hair matrix and outer root sheath of hair follicle. Why not having look at the expression of these growth factors in vivo to show which pathway(s) is/are modulated by GPHD?

Reply: We indeed investigated the level of growth factors in vivo and the results are shown in figure 1D.

12) In the Figure 3B, what is GPE? Is it GPHD?

Reply: Correction is made at figure 3B.

13) In the summary diagram, the authors assumed that the growth factors are secreted to the medium and act as growth factors for hDPC. This claim is too strong based on the study results, especially the fact that these growth factors act via an autocrine loop. To make this claim, the authors would need at least to perform experiments with neutralizing antibodies.

Reply: We completely agree with this point. In fact, we only measured the amount of mRNA for these growth hormones, not whether they are secreted out of the cell or whether they act through an autocrine loop. Therefore, we have added a dashed line to Figure 6 to indicate that this is not experimentally proven, but only the authors' speculation.

14) In the discussion the authors say that the expression of IGF-1 has been reported to stimulate the hair elongation and promote the anagen phase in of dermal papilla cells [44]. The reference cited here does not show this claim and IGF-1 is the most potent anagen prolonging growth factor in hair follicle but not in dermal papilla. Please revised this sentence in the discussion.

Reply: We revised this sentence at line 367. “Secretion of IGF-1 from dermal papilla cells has been reported to stimulate hair follicle growth via PI-3 kinase pathway.” We further replaced the reference with a new one that directly supports this fact (ref 50).

Br J Dermatol. 2009 Jun;160(6):1157-62. doi: 10.1111/j.1365-2133.2009.09108.x. Epub 2009 Mar 26.l-Ascorbic acid 2-phosphate promotes elongation of hair shafts via the secretion of insulin-like growth factor-1 from dermal papilla cells through phosphatidylinositol 3-kinase

15) as they are a known an instructive niche for progenitor and epithelial stem cells in their development and Regeneration [35]: Apart the fact that this sentence is absolutely similar to the title of the reference 35, Isn't the source of the hair follicle epithelium located mainly in the bulge?

Reply: We believe this point is of similar importance to question #13 and have revised the sentence at line 376 by deleting the part where the author expects growth hormone to act in an autocrine and paracrine manner as well as an indicated similar expression. We have also deleted reference 35.

Comments on the Quality of English Language

Some grammar mistakes need to be corrected. See a few examples below:

1) therapies drugs : it should be either drug therapies or therapeutical drugs.

2) an emergence of safer alternatives is rising: the emergence of safer alternative is required and start to rise

3) Similar strategy was followed in present study: Similar strategy was followed in the present study

4) anti-obesity, and anti-anxiety effects, not much is known about: anti-obesity, and anti-anxiety effects [21, 22], but not much is known about

5) as they are a known an instructive niche: as they are a known instructive niche

6) the hair elongation and promote the anagen phase in of dermal papilla cells: the hair elongation and promote the anagen phase of hair follicles

Reply: The indicated errors are all corrected.

Reviewer 2 Report

Comments and Suggestions for Authors

The manuscript submitted for evaluation concerns research on products that can complement the treatment of baldness. The problem is quite important, especially nowadays, when most people struggle with ubiquitous stress, an important factor causing this disease. Unfortunately, the used drugs show many side effects. Therefore, the search for alternative substances effective in such therapy is necessary. The described research focuses on an attempt to explain the properties and pathways of action of Gynostemma pentaphyllum (GPHD) leaf extracts. The authors concluded that damulin B (the main component of the plant) showed hair growth-promoting properties in vitro. They indicated that minoxidil, GPHD, and damulin B induce hair growth through the Wnt/β-catenin pathway via AKT signaling, which is confirmed by in vivo studies. It seems that the presented research and its further continuation may provide a product that will effectively support the therapy of people suffering from hair loss.

Minor comments:

The order of Refs. [24] and [25] should be changed, in accordance with the order in which these papers are cited in the text.

The title "Conclusion" should be capitalized.

All authors should be included in the cited articles, instead of the abbreviation "et al.".

Author Response

Reviewer 2

Thank you very much for your positive review of my manuscript. I have addressed all of the points you raised and hope that these responses will be to your satisfaction.

The manuscript submitted for evaluation concerns research on products that can complement the treatment of baldness. The problem is quite important, especially nowadays, when most people struggle with ubiquitous stress, an important factor causing this disease. Unfortunately, the used drugs show many side effects. Therefore, the search for alternative substances effective in such therapy is necessary. The described research focuses on an attempt to explain the properties and pathways of action of Gynostemma pentaphyllum (GPHD) leaf extracts. The authors concluded that damulin B (the main component of the plant) showed hair growth-promoting properties in vitro. They indicated that minoxidil, GPHD, and damulin B induce hair growth through the Wnt/β-catenin pathway via AKT signaling, which is confirmed by in vivo studies. It seems that the presented research and its further continuation may provide a product that will effectively support the therapy of people suffering from hair loss.

Minor comments:

The order of Refs. [24] and [25] should be changed, in accordance with the order in which these papers are cited in the text.

Reply: We have carefully checked the order of these references and found that there are no errors. 

The title "Conclusion" should be capitalized.

Reply: Correction is made

All authors should be included in the cited articles, instead of the abbreviation "et al.".

Reply: The citation format will be reorganized according to the editor’s instructions.

Round 2

Reviewer 1 Report

Comments and Suggestions for Authors

Thanks to the authors for replying to my comments. However, I still have a few points that still need clarification: 

1) Cyclosporin is not used only for alopecia areata but also for androgenetic alopecia. Please add some information about it in the introduction. 

2) Concerning the role of DP cells in hair cycle, please also add that other cell types are involved in the regulation of hair cycle with corresponding references.

3) DP cells cell line often do not retain their inductivity properties and also can lose them over passaging. It is important to show at least expression level of inductivity marker, either versican or alkaline phosphatase, especially if the authors can't perform the spheroid experiment.

4) Concerning the mode of administration of GPHD, it would be great to add the explanation given in the reply to my previous comment in the material and methods. Also, mixing with some PEG6000 could help to get enough viscosity to apply topically. 

5) I understand that the authors want to go more in depth in a follow up study the mechanisms of hair growth promotion but it is important in the present study to get the quantitative hair cycle staging to be convincing. 

6) About looking at the expression of the growth factors in vivo, I wasn't clear, sorry. I want to say that it would be better to have quantitative data at the protein level (by immunofluorescence or immunohistochemistry) to show that the difference of expression seen at the mRNA level reflect changes in the hair follicle and not anywhere else in the skin.

Comments on the Quality of English Language

Thanks for correcting the different points mentioned previously.

Author Response

Reviewer 1:

Thanks to the authors for replying to my comments. However, I still have a few points that still need clarification: 

Reply: I would like to express my deepest gratitude to the reviewer for helping to improve my research significantly.

 1) Cyclosporin is not used only for alopecia areata but also for androgenetic alopecia. Please add some information about it in the introduction. 

Reply: added a description of cyclosporine and two new references (13, 14) on lines 54-57 of the introduction   

“Furthermore, cyclosporine is frequently employed in alopecia areata treatment, either as a standalone therapy or in combination with other treatments. It is also utilized alongside certain antioxidants and immunosuppressants for managing androgenic alopecia. However, the mechanism of action of this drug is yet to be elucidated.”

Ref 13: Palakkal, S., et al., Effect of cyclosporine A - tempol topical gel for the treatment of alopecia and anti-inflammatory disorders. Int J Pharm, 2023. 642: p. 123121.

Ref 14: Nowaczyk, J., et al., Cyclosporine With and Without Systemic Corticosteroids in Treatment of Alopecia Areata: A Systematic Review. Dermatol Ther (Heidelb), 2020. 10(3): p. 387-399.

2) Concerning the role of DP cells in hair cycle, please also add that other cell types are involved in the regulation of hair cycle with corresponding references.

Reply: added the following description along with two new references (46, 47) on line 335-341 of the Discussion.

“The hair follicle contains diverse cell types essential for its growth and structure. Hair matrix cells, situated close to the dermal papilla, interact with it to stimulate the formation of epithelial elements like the inner root sheath and medulla. The hair shaft, primarily composed of keratinocytes, and the inner and outer sheaths collectively constitute the epidermal components of the hair follicle. The dermal papilla supplies nutrients for hair growth and follicle maintenance, forming a multicellular tissue structure crucial for inducing hair growth.”

Ref 46: Ge, W., et al., Single-cell Transcriptome Profiling reveals Dermal and Epithelial cell fate decisions during Embryonic Hair Follicle Development. Theranostics, 2020. 10(17): p. 7581-7598.

Ref 47:Lin, X., L. Zhu, and J. He, Morphogenesis, Growth Cycle and Molecular Regulation of Hair Follicles. Front Cell Dev Biol, 2022. 10: p. 899095.

3) DP cells cell line often does not retain their inductivity properties and also can lose them over passaging. It is important to show at least the expression level of inductivity marker, either versican or alkaline phosphatase, especially if the authors can't perform the spheroid experiment.

Reply: As the reviewer pointed out, we investigated ALP and versican expression by performing RT-PCR. We also compared it with the primary cells (passage No. 2 & 8) and found that it was expressed in significant amounts.

Moreover, recent studies have demonstrated that the DP cell line used in our study expresses alkaline phosphatase, which is increased to the same extent as primary male and female DP cells in response to L-fructose and marine-derived sugars (Ref 33). The cell line also expresses DPC markers such as alpha-smooth muscle actin and biglycan, and is responsive to the Wnt-beta-catenin and BMP signaling pathways, which are required for the hair-inducing activity of DPC (Ref 32). These results suggest that this cell line retains the properties of primary DPCs. We have added the following reference showing the expression of alkaline phosphatase at line 104.

Ref 33: Effect of Marine-Derived Saccharides on Human Skin Fibroblasts and Dermal Papilla Cells.Augustyniak A, McMahon H.Mar Drugs. 2023 May 27;21(6):330. doi: 10.3390/md21060330.

4) Concerning the mode of administration of GPHD, it would be great to add the explanation given in the reply to my previous comment in the material and methods. Also, mixing with some PEG6000 could help to get enough viscosity to apply topically. 

Reply: added the following description on lines 146-153 of Materials and Methods.

As previously stated, determining the concentration of GPHD is challenging compared to minoxidil due to its extract nature, and applying it in ointment form would further dilute the concentration due to viscosity. Even if applied topically, we anticipated that its effect wouldn't match that of minoxidil, requiring more frequent applications, which would disrupt methodological uniformity. Moreover, since the extract showed no toxicity when administered orally, priority was given to oral administration to ascertain the volume of GPHD at least.

5) I understand that the authors want to go more in depth in a follow up study the mechanisms of hair growth promotion, but it is important in the present study to get the quantitative hair cycle staging to be convincing. 

Reply: As suggested by the reviewer, we have analyzed the hair cycle stages following the methodology described in Reference 36 and have added the results as a bar graph below Figure 1C. We have also added a new description of the methodology (Section 2.7) and references. We have added a description of the results in the results (line 225-228).

6) About looking at the expression of the growth factors in vivo, I wasn't clear, sorry. I want to say that it would be better to have quantitative data at the protein level (by immunofluorescence or immunohistochemistry) to show that the difference of expression seen at the mRNA level reflect changes in the hair follicle and not anywhere else in the skin.

Reply: The main objective of this study was to determine the hair growth effects of an extract from the leaves of GPHD. In the future, we plan to further study the detailed mechanisms and the potential for GPHD to be developed as a treatment for hair loss. We are also investigating whether GPDH can prevent stress-induced hair loss. In the process, we will study in detail the expression levels of various growth factors, beta-catenin, AKT, BMP4, ALP, etc. in the mouse skin tissue by immunofluorescence. Therefore, we kindly request permission to publish the results pointed out by the reviewer in the following paper.
